

# Intensity of geodynamic processes in the Lithuanian part of the Curonian Spit

Algimantas Česnulevičius[1], Regina Morkūnaitė[2], Artūras Bautrėnas[1], Linas Bevainis[1], Donatas Ovodas[1]

[1] Centre for Cartography, Faculty of Chemistry and Geosciences, Vilnius University, LT-03101, Vilnius, Lithuania.

[2] Labaratory of Quaternary Research, Institute of Geology and Geography, Nature Research Centre, LT-08412, Vilnius, Lithuania.

Correspondence author Algimantas Česnulevičius, email: algimantas.cesnulevicius@gf.vu.lt

**Abstract.** The paper considers conditions and intensity of aeolian and dune slope transformation processes occurring in the wind-blown sand strips of the dunes of the Curornian Spit. An assessment of the intensity of aeolian processes was made based on the analysis of climatic factors and in-situ observations. Transformations in aeolian relief forms were investigated based on the comparison of

geodetic measurements and measurements of aerial photographs. Changes in micro-terraces of dune slopes were investigated through comparison of the results of repeated levelling and measurements of aerial photographs. The periods of weak, medium and strong winds were distinguished, and sand moisture fluctuations affecting the beginning of aeolian processes were investigated. The wind-blown sand movements were found to start when sand moisture decreased by 2 % in the surface sand layer and by up

to 5 % at a depth of 10 cm. In 2004 – 2016, the wind-blown sand movements affected the size of reference deflation relief forms: scarp length by 8 %, scarp width by 35 %, pothole length by 80 %, pothole width by 80 %, roll length by 17 %, roll width by 18 %, hollow length by 17 %, and hollow width by 39 %. The elementary relief forms in the leeward eastern slopes of the dunes experienced the most intensive transformations. During a period of five months, the height of micro-terraces of the eastern slope of the

Parnidis Dune changed from 0.05 to 0.64 cm. The change was related with fluctuations in precipitation intensity: in July – August 2016 the amount of precipitation increased 1.6-fold compared with the multiannual average, thus causing the change in the position of terrace ledges by 21 %.

References 24, figures 7, tables 3.

**Keywords**: aeolian processes, wind, precipitation, sand moisture, micro-terrace, marl, unmanned
aerial vehicle

## 1. Introduction

The formation of the core of the Curonian Spit started 8–6 thousand years ago and is related with
the end of the third transgression of the Littorina (L3, III bl) Sea. The further development of the spit took place during the Post-Littorina L 4 (IV PL) period (Bitinas et al., 2001; Bitinas and Damušytė, 2004).





Southward transportation of sand formed a shallow littoral zone on the south-eastern coast of the Baltic Sea. In the zone, fine- and medium-grained sand was drifted against glacigenic hills and onto emerged sand, grain and pebble shallows of marine origin (Bitinas and Damušytė, 2004; Bitinas et al., 2005) (Fig. 1).


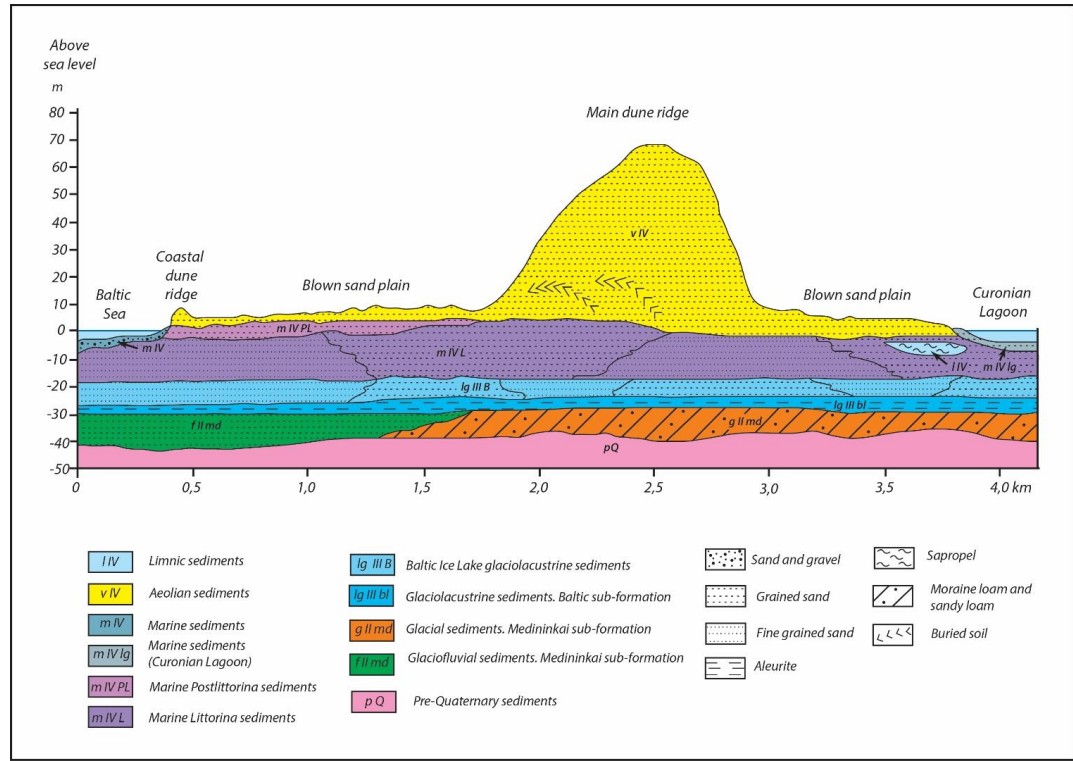

Fig. 1. Cross-section in the northern part of the Curonian Spit (Vingiakopė environment) (by Badyukova et al., 2007; Bitinas et al., 2002).

45         Favourable conditions for the formation of the spit and its dunes occurred before the start of the third stage of transgression of the Littorina Sea, when water level on the south-eastern coast was 6–7 m lower compared to the today's water level. With water level rising, the Semba Peninsula suffered intensive erosion and sediments were transported northwards. The germ of the spit formed near Šarkuva settlement (phase I); it reached Rasytė Island and grew further (phase II). 4–4.5 thousand years ago, the spit extended

up to Juodkrantė settlement (phase III), and 2 thousand years ago, it reached the continental coast (stage IV) (Gudelis, 1979, 1998; Kabailene 1967; Kunskas, 1970; Kliewe and Janke, 1982; Mojski, 1988; Müller, 2004; Starkel, 1977) (Fig. 2).

        Wind-blown sand movement is the main geomorphological geodynamic process that currently occurs in the Curonian Spit. Since the 16th century, 14 villages have been entirely buried under the sand,

and dwellers of other villages had to move from one place to another to escape wind-blown sand (Fig. 3).



The first efforts to slow down wind-blown sand movement go back to the first half of the 19th century when dunes started to be overgrown with vegetation, which stabilized intensive movement of sand throughout the 19th century. The first scientific research into aeolodynamic processes in the Curonian Spit appeared in the second half of the 19th century (Berendt, 1869). Since the middle of the 20th century,

the Curonian Spit has become an object of comprehensive investigations by Lithuanian geographers who put their main focus on the morphogenesis of dunes and the mineral and granulometric composition of sand (Gaigalas, Pazdur, 2008; Gudelis, 1998; Mardosienė, 1988; Michaliukaitė, 1967; Minkevičius, 1982; Minkevičius et al., 1996). Though by the end of the 20th century only four segments of wind-blown dunes were left in the Curonian Spit, a dramatic increase in the number of visitors to the dunes not overgrown

with vegetation brought about significant relief transformations. At the initiative of the Curonian Spit National Park, since 2003 the monitoring of aeolian processes has been pursued in the Lithuanian part of the Curonian Spit, which enables identifying regularities of changes in aeolian deflation and accumulation processes (Morkūnaitė, Česnulevičius, 2005; Česnulevičius et al., 2016).

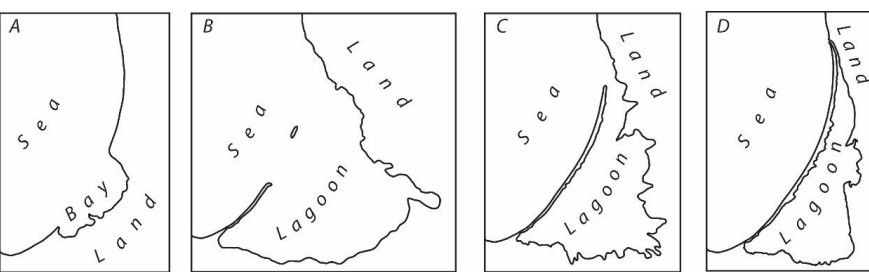


Fig. 2 Curonian Spit evolution phases: A – beginning of the third stage of Littorina Sea transgression (4ky BC), B – end of Littorina Sea transgression (3 ky BC), C – Mya Sea transgression (0,4 kY BC), D – contemporary Baltic Sea and Curonian Lagoon (Aufgenommen vom … 1880; Gudelis, 1998; Kabailiene, 1967; Kunskas, 1970).


Currently, specific tectonic subsidence phenomena related to deformations of marl accumulated in the freshwater basin are taking place south of Nida settlement (Fig. 4). Due to this reason, marl squeeze formed on the coast of the Curonian Lagoon. Before the Second World War, Vladas Viliamas described the phenomena and pointed out a rather wide distribution of the phenomena south of Juodkrantė

settlement (Viliamas, 1932). In 1985, marl squeeze was still observed south of Juodkrantė (Kabailienė, 1997). To date, marl can only be detected south of Nida. The pressure of a sand layer on the plastic marl layers causes deformations on the surface of dunes and gives rise to the formation of stepwise micro-terraces. Such geological-geomorphological formations can also be found in other areas of the Baltic Sea (Lampe et al., 2011; Sergeev et al. 2016).





The purpose of this article is to assess quantitative changes of aeolian relief forms induced by alterations of climate components in the wind-blown areas of the Curonian Spit.

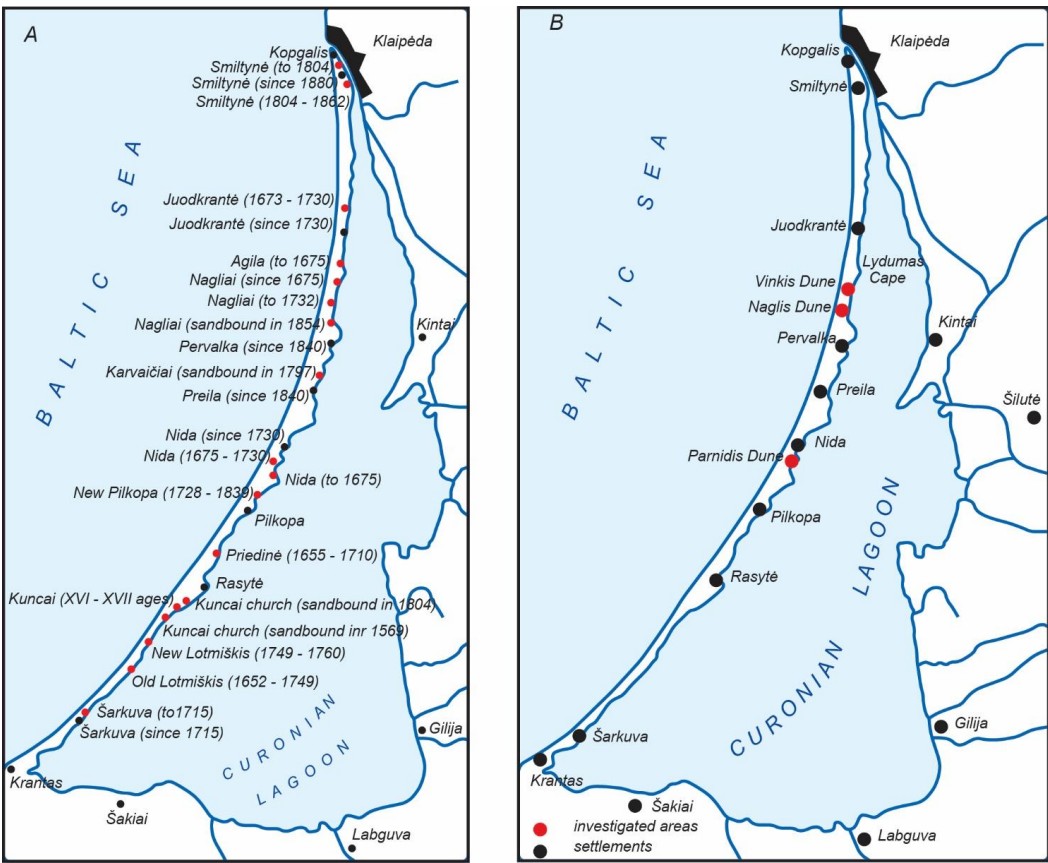

Fig. 3. Contemporary and extinct settlements in the Curonian Spit (A) and investigated areas (B).

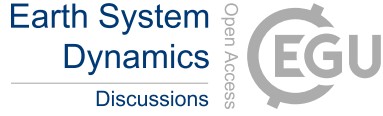



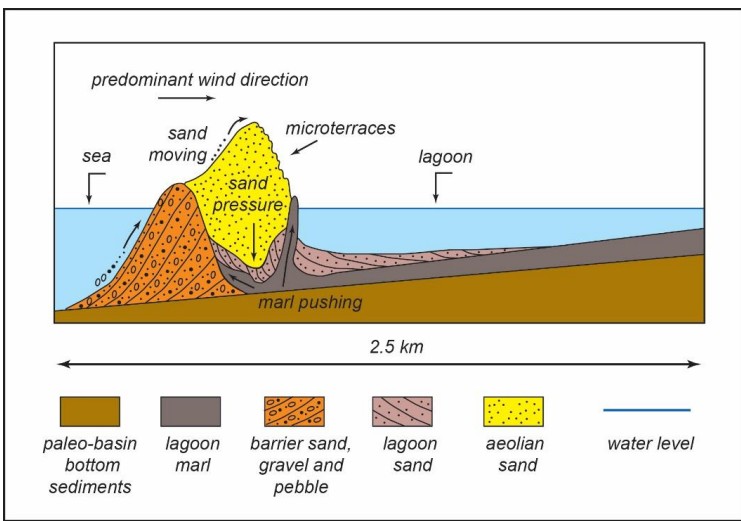

Fig. 4. Lagoon marl layer deformation model in the investigated area (by Sergeev et al., 2016.)

## 2. Methods


The intensity of aeolodynamic processes was assessed using repeated precise levellings in the reference strips of wind-blown dunes and analysing the digital topographical large-scale maps (1:5 000) of 2004, topographical maps of 1910 and 1940, orthophoto images of 2009 and 2014, and aerial

photographs taken from an unmanned aerial vehicle in 2015–2016. Investigations were carried out in the Lithuanian part of the Curonian Spit: Parnidis, Naglis, and Vinkis dunes (Fig. 3 B), where a repeated precise levelling of wind-blown sand strips of the ridge was performed. The levelling was done with the help of an electronic tacheometer ELTA 500 and GPS Trimble GeoXT 6000 with an accuracy of 5 ″ for measuring angles and 1 mm for measuring elevations of points. A comparison of repeated levelling profile

data enabled quantitative assessment of changes in the Main Dune Ridge relief forms. The mapping of the surface of dunes was done using an unmanned aerial vehicle DJI Inspire.

An analysis of aeolian microform variations (depressions, hummocks, gullies, ridges) was carried out in a wind-blown sand strip of Naglis and Vinkis dunes. Quantitative changes of microforms that occurred in the period of 1999–2016 were compared. The morphometric microform indices were obtained

by performing route measurements from natural benchmarks and using a GPS receiver and a ±1 mm accuracy distance measurer Leica Disto D510.

An assessment of dune surface changes was made based on meteorological factors: the beginning and end of strong wind periods and the moisture of the surface sand layer. An analysis of sand samples taken from the surface sand layer and at a depth of 0.15 m in windward and leeward dune slopes enabled

determining the beginning of local and massive wind-blown sand movements. Sand moisture and groundwater level in the dunes were estimated using georadar scanning data (Dobrotin et al., 2013).



To specify in greater detail the periodicity and intensity of wind-blown sand movements, an analysis of climatic parameters (wind speed, wind direction, air temperature, and precipitation) from meteorological stations of Klaipėda, Nida and Šilutėų for the period 1991–2015 was performed: In addition, the data from a temporary meteorological station established on the seacoast of Nida for the summer of 2016 were used. The assessment of climatic parameters was necessary for the purpose of forecasting possible meteorological situations. Unfavourable weather conditions (wind, rain, snow) made a direct impact on the flight of an unmanned aerial vehicle and the quality of aerial photographs.

## 3.Climatic factors

An analysis of thirty-year data of climatic factors from meteorological stations of Nida, Klaipėda, and Šilutė enabled distinguishing the most intensive periods of aeolian processes. In particular, three main factors that made a direct impact on aeolian processes, i.e. wind regime, precipitation, and air temperature, were analysed.

Compared with the remaining territory of Lithuania, the Curonian Spit distinguishes by a specific climate with mild and often snowless winters, frequent strong winds and storms, a longer period of above-zero temperatures. We described the climate of the northern part of the Curonian Spit based on the data from the Klaipėda meteorological station and the climate of the central part of the spit based on the data from Nida and Šilutė meteorological stations.

Air temperature was an important factor affecting aeolian processes. In the cold season of the year, aeolian processes were very slow and occurred only with storm winds. Due to low temperature, the surface of moist sand got frozen and the crust formed, which protected sand from being blown by wind. The cold season of the year lasted from 18 (1991 –1992) to 114 (1995 – 1996) days. Cold seasons were often accompanied by thaw periods when the frozen sand crust lost its stiffness and aeolian processes revived. The thawing of the sand crust required longer (4 – 16 days) periods of temperatures above zero (1 – 4 C°). Such situations were recorded in 1986 – 1987, 1989 – 1990, 1992 – 1993, 1999 – 2000, and 2008 –2009. In 1991 – 1992, when temperatures above zero prevailed, no sand crust formed and aeolian processes could take place all year round.

The amount and temporal distribution of precipitation were other important factors affecting aeolian processes, in particular, the number and duration of periods without precipitation. An analysis of thirty-year data of precipitation from Nida, Klaipėda and Šilutė meteorological stations showed that dry periods were distributed rather evenly throughout a year. Dry periods without precipitation lasted for 15 days in spring (March – May), 13 days in summer (June – August), and 10 days in autumn (September – November), on average. However, dry periods differed between years. For example, the period without



precipitation lasted 30 days (entire May month) in the spring of 1992, 25 days (nearly all the month of May) in the spring of 2000, and 26 days (nearly all the month of April) in 2009. In spring, the minimum duration of the period without precipitation was 6 days, and the maximum 30 days. In summer, the longest

period without precipitation lasted 29 days (July 1994), and the shortest 5 days. In autumn, the longest periods without precipitation were recorded in 1998 (21 days in September – October) and 2005 (19 days in October).

   Wind was the main climatic factor affecting aeolian processes. Wind speed, direction and duration of blowing varied between different seasons of the year (Fig. 5). An analysis of multiannual

weather parameters from the Nida meteorological station showed that the Curonian Lagoon made a great impact on wind dynamics. E-SE-S winds of the average speed of 5 – 6 m/s prevailed in all years. According to the data from the Nida station, the maximum speed of winds lasting three hours and longer reached 16 m/s in spring, 16 m/s in summer, 22 m/s in autumn, and 21 m/s in winter. According to the data from the Klaipėda seacoast meteorological station, SW – W – NW winds of the averaged speed of

5.0 m/s prevailed.  The maximum speed of three-hour winds taken at the Klaipėda hydrometeorological station was considerably greater: 20 m/s in spring (2000), 18 m/s in summer (1981 and 2002), 26 m/s in autumn (1981), and 26 m/s in winter (1982). Among maximum wind speeds, N (51 % of cases), SW (38 %), and SE (11 %) winds prevailed. The maximum wind speeds of other directions constituted less than 1 %. According to the data from the Šilutė meteorological station, the maximum wind speed reached

17 m/s in spring, 13 m/s in summer, 21 m/s in autumn, and 18 m/s in winter.



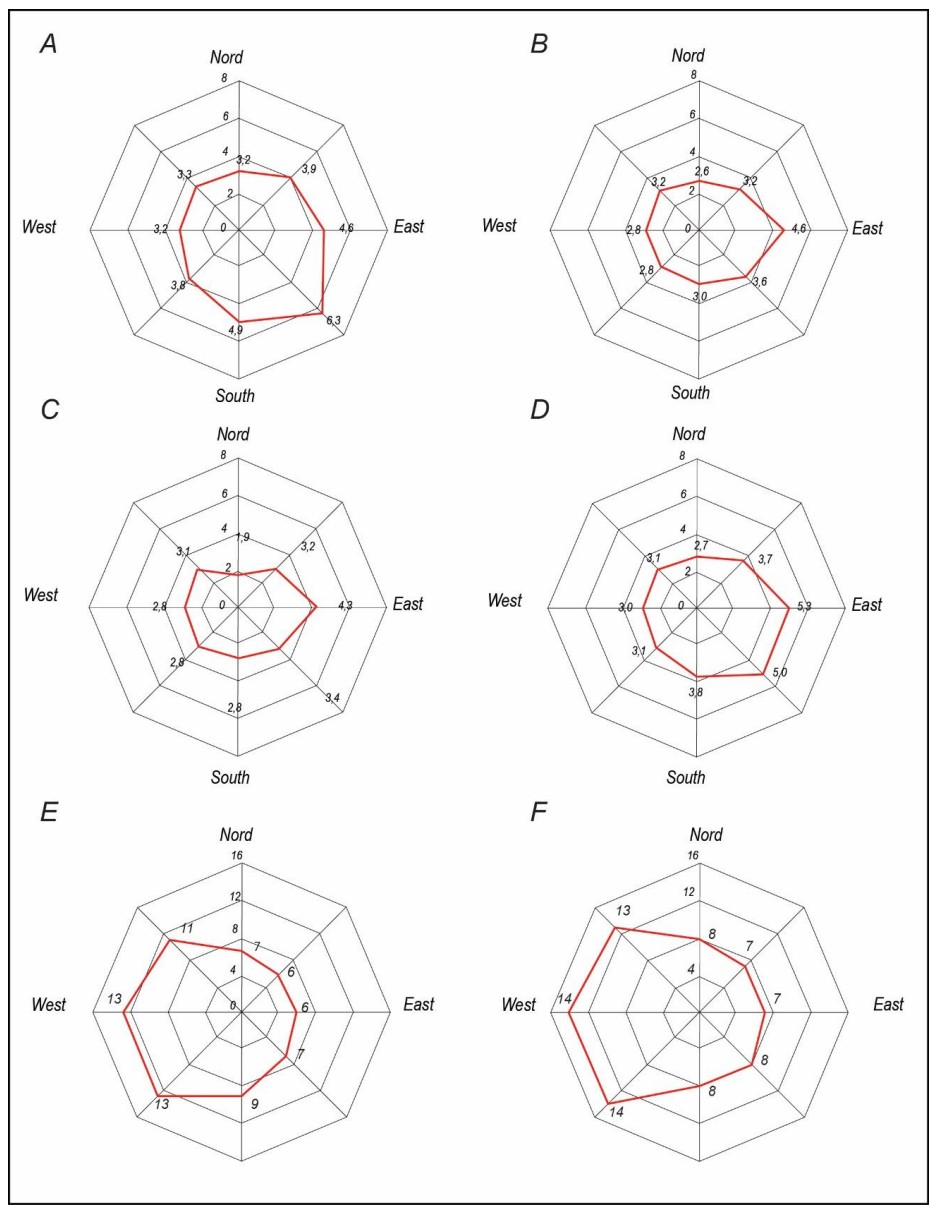

Fig. 5. Multiannual average wind speed registered at meteorological stations in Nida, Šilutė
and Klaipėda: A – December – February (Nida); B – March – May (Nida); C – June –August (Nida); D
– September – November (Nida); E - multiannual maximum wind speed (Šilutė); F – multiannual
maximum wind speed (Klaipėda).


The recorded differences in wind regime were due to the location of meteorological stations:
Klaipėda and Šilutė hydrometeorological stations were exposed to the winds of all directions, whereas



the Nida meteorological station was located at the foot of the dune ridge of the Curonian Spit, and the dune ridge blocked EW – W – NW winds.

To measure the local wind regime in the coastal area of the Baltic Sea, a temporary mobile meteorological station was established. It operated only in the summer of 2016. The data about wind speed, wind direction, air temperature, and precipitation taken by self-recorders of the station were

compared with the data from the Nida meteorological station, and correlation indices were determined. Occasional wind speed measurements were also taken by an unmanned aerial vehicle. A comparison of short-term and occasional measurements of wind speed and direction with the data of meteorological stations allows stating that the wind regime recorded at the Klaipėda meteorological station was closest to the wind regime recorded in the dune ridge of the Curonian Spit. The correlation coefficient was equal

to r = 0.988 for wind speed and  r = 0.998 for wind direction.

## 4. Results and discussion

Our analysis was based on the data of wind speeds exceeding 6 m/s as recorded at Nida, Klaipėda

and Šilutė meteorological stations in the warm periods (March – October) of 1984–2015. During the investigation period, W-NW (16.8 %), W (14.0 %), S (3.7 %), and E (2.7 %) winds prevailed among wind speeds over 6 m/s, which constituted 13.7 % of all cases. The Klaipėda meteorological station recorded a total of 55 cases of very strong winds (over 15 m/s) in 2006 – 2015 (a shorter period of strong winds), 75 cases of very strong winds in 1984 – 1994 (a long period of very strong winds), and 63 cases in 1995 –

2005 (a medium-duration period of very strong winds). In 2006 – 2015, 60 cases of very strong winds were recorded at the Nida station and 43 cases at the Šilutė station. The warm period of 2016 in the Curonian Spit was particularly windy. According to the data from the Nida meteorological station (wind speed and direction were measured every 3 hours), winds were recorded to blow for a total 4119 hours, and winds stronger than 6 m/s were recorded to blow 3966 hours (Table 1) during the warm season.

Permanent strong winds caused fast sand drying and intensified deflation processes. Our measurements showed that one hour after rain the surface sand layer was about 50 per cent drier compared with sand at a depth of 10 cm (Table 2).

Sand moisture was measured by the method of weighing dishes in different dune profile areas, and a comparison was made between sand moisture in the Parnidis Dune environs of 16 June 2016

(amount of precipitation 2.3 mm) and in the Vinkis Dune environs of 18 June of the same year (amount of precipitation 1.5 mm) (Table 2). In the western part of the Vinkis Dune, sand moisture reached 1.91– 2.74% on the surface and 1.83–2.39% at a depth of 10 cm. In the nature trail, sand moisture was 0.22% on the surface and 0.065 at a depth of 10 cm. Such inverse distribution could be due to human trampling. In May 2014, no moisture distribution inversion was detected on sand surface or in deeper layers: on 12



May (amount of precipitation 2.1 mm) sand moisture reached 3.41% on the surface and 5.04% at a depth of 15 cm near the nature trail up the Nagliai Dune (less visitors). The sand moisture measurement data obtained near the Parnidis Dune on 27 June 2015 (amount of precipitation 1.9) were more consistent, because in the western part of the slope there was a tussock area where sand contained more moisture (by 1–2 %) compared to the sand near the top.

Table 1. Wind speed and number of cases in March–October 2016 (data from the Nida meteorological station).

| Wind direction | Cases | | Total winds | | | Average wind speed over 6 m/s | |
|---|---|---|---|---|---|---|---|
| | Number | % | Maximum speed in gust, m/s | Minimum speed, m/s | Duration, in hours | Number | % |
| N | 273 | 20 | 24.7 | 0.0 | 819 | 259 | 95 |
| NE | 117 | 9 | 24.7 | 0.0 | 819 | 109 | 92 |
| E | 104 | 8 | 24.3 | 1.0 | 354 | 99 | 94 |
| SE | 81 | 6 | 25.5 | 0.0 | 315 | 81 | 100 |
| S | 99 | 7 | 26.4 | 1.0 | 243 | 95 | 96 |
| SW | 213 | 15 | 29.8 | 1.0 | 297 | 212 | 99 |
| W | 245 | 18 | 26.6 | 1.0 | 639 | 239 | 98 |
| NW | 239 | 17 | 26.5 | 1.0 | 735 | 228 | 95 |

Table 2. Distribution of sand moisture on the surface and at a depth of 10 cm in the environs of Parnidis and Naglis dunes.

| Location | Coordinates, m | Moisture, % | | Precipitation, mm |
|---|---|---|---|---|
| | | On the surface | 10 cm deep | |
| Parnidis Dune | | | | |
| Western slope foot | X = 6133110 Y = 3086571 | 2.24 | 5.01 | 2.3 |
| Western slope | X = 6132967 Y = 3087035 | 5.74 | 4.02 | 2.3 |
| Naglis Dune | | | | |
| Western slope foot | X = 6150431 Y = 315518 | 1.84 | 3.93 | 1.5 |
| Western slope | X = 6150432 Y = 315788 | 2.38 | 1.96 | 1.5 |
| Top | X = 6150463 Y = 316129 | 0.25 | 0.45 | 1.5 |
| Eastern slope | X = 6150433 Y = 316441 | 1.61 | 0.15 | 1.5 |
| Eastern slope foot | X = 6150459 Y = 316492 | 1.53 | 2.91 | 1.5 |

Because of an intensive stream of visitors, the crest of the southern segment of the dune ridge of the Lithuanian part of the Curonian Spit (Parnidžis–Sklandytojai dunes) is completely trampled. Small



natural deflation hollows have survived only in the distal slope facing the Curonian Lagoon. Such hollows

are most intensively affected by S – SE winds prevailing in spring (May) and winter (December – February). S – SE winds in the summers of 2004 – 2016 markedly changed the form and depth of deflation hollows. The changes constituted up to 0.05 – 0.20 m per day.

The relief microforms reflecting the intensity of short-term deflation processes have formed in nearly all investigated deflation and accumulation hollows in the northern Juodkrantė – Pervalka segment.

Scraps (up to 2 m height and up to 40° inclination), deflation mini-gullies, potholes, accumulative steps, small ridges and rolls formed in the hollows. Deflation hollows were found to be mostly distributed between the Lydumo ragas peninsular and Vinkis Dune. The deflation hollows were 35 – 40 m a.s.l., which shows that the best conditions for deflation were on the top of the dune ridge intensively affected by W winds.

Deflation hollows formed in each seaward or lagoonward relief depression in the Lydumo ragas – Vinkis Dune segment. We compared the measurements of 1999 – 2016 and found that the length and width of deflation hollows were the most variable parameters. Besides, the length and width of passages connecting the hollows were also very dynamic. They could change by some to several dozens of meters per year (Table 3). In the Juodkrantė – Pervalka segment, the blowing of sand away from hollows was

due not only to frequent strong winds, but also to the position of hollows in dune slopes. The most intensive deflation processes took place in the hollows located in the leeward eastern slopes of the dune ridge. The hollows in the crest of the dune ridge were blown out at a lower degree, and the hollows in the windward western slope of the dune ridge suffered the lowest degree of deflation. The degree of deflation also depended on whether hollows were open or closed. The hollows with W-E deflation passages

opening thereto suffered the fastest transformations. The wind quickly transformed such hollows into deflation gullies.

Table 3. Transformation of aeolian deflation relief forms in 1999 – 2015.

| Aeolian form | Length/width, m | | | |
|---|---|---|---|---|
| | 1999 | 2001 | 2003 | 2016 |
| Scarp | 6.6 / 2.0 | 7.1 / 2.4 | 6.6 / 1.7 | 6.8 / 1.8 |
| Hollow | 30.3 / 13.2 | 34.6 / 16.4 | 28.7 / 14.8 | 35.6 / 18.3 |
| Passage | 30.0 / 4.0 | 42.0 / 7.6 | 34.0 / 6.3 | 47.0 / 11.8 |
| Roll | 29.0 / 5.0 | 30.5 / 5.5 | 31.0 / 4.8 | 34.0 / 5.9 |
| Ledge | 15.4 / 8.5 | 17.2 / 7.6 | 19.6 / 9.4 | 22.6 / 11.4 |
| Pothole | 25.6 / 8.2 | 39.3 / 11.2 | 42. 0 / 13.8 | 46.2 / 15.6 |


Further to the south, the Parnidis Dune sand layer is pressing a freshwater marl layer beneath and forming an up-to 2 m high squeezed scrap on the coast of the Curonian Lagoon. The pressure causes the formation of micro-terraces on the slope of the dune. The development of micro-terraces is closely related with infiltration of precipitation into the sand layer. The pressure of the sand layer increases in spring




when snow is melting and in summer and autumn when precipitation is abundant. This causes changes in morphometric parameters of micro-terraces on a stable slope of the dune ridge south of the Parnidis Dune. In April and September of 2015 – 2016, the slope was photographed 4 times from an unmanned aerial vehicle. Upon creation of a three-dimensional surface model, the relief microforms were measured. The measurements were taken in the photographs of different periods where the spatial position of the same

points was precisely identified (Fig. 6).

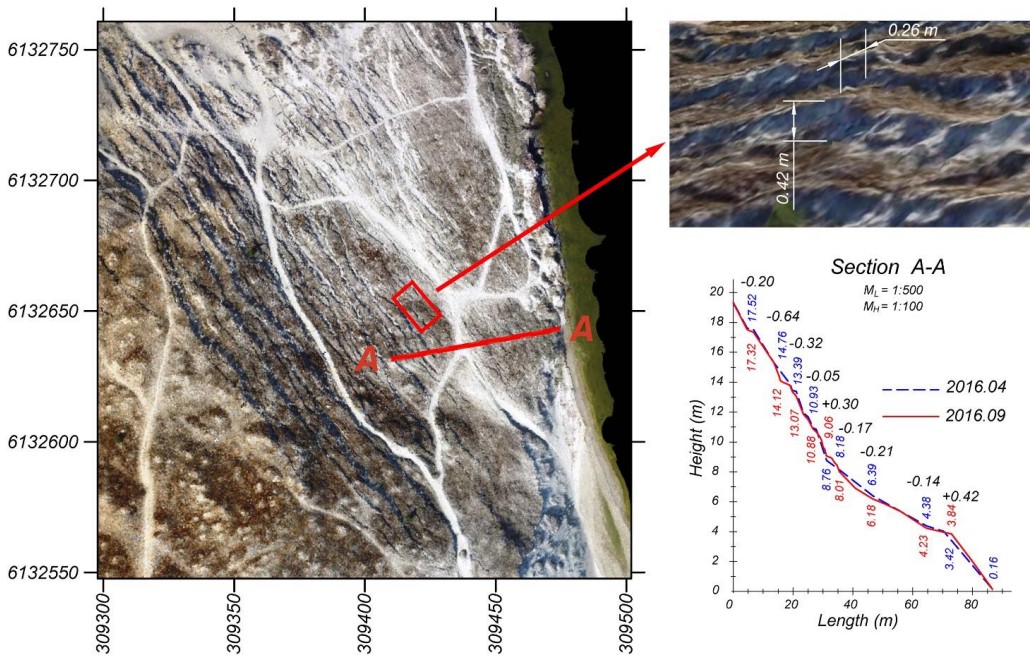

Fig. 6. Investigation areas (left) and changes in micro-terrace parameters (right).

The measurements showed that the position of micro-terraces changed during a five-month period. A comparison of the measurement results of aerial photographs between April and September of 2016 revealed a decline in the height of terrace ledges by 05 – 0.64 cm. An increase in the height of the terrace ledge in the middle part of the slope was due to a sand slide made by numerous visitors to the dunes.

     The height of the coastal scrap at the foot of the slope increased by 0.42 m thanks to the marl

squeeze. Similar changes occurred on the eastern slope of the Sklandytojai Dune located on the Lithuanian – Russian border where the height of the coastal marl scrap increased by 0.5 m during the spring – summer period of 2016.

     Rather sharp changes in micro-terraces of eastern slopes were related with the amount of precipitation exceeding the average. In April–September of 2016, precipitation amount exceeded the





multiannual average by 110%. Still greater differences were observed in July–August of 2016 when the

amount of precipitation exceeded the multiannual average by 160% (Fig. 7).

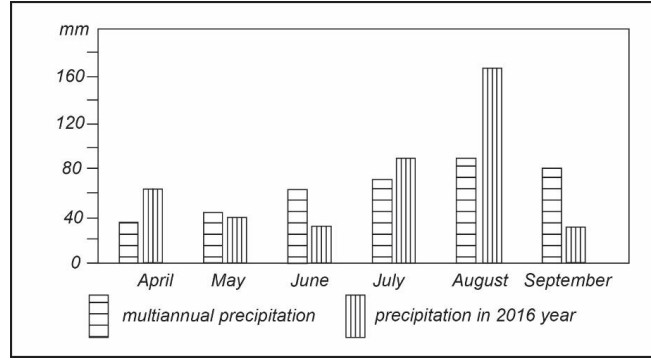

Fig. 7. Comparison of multiannual and annual (2016 year) summer precipitation (by Nida
285                                     meteorological data).

We can state that in the end of summer the pressure of moist sand on the marl layer increased 1.6-

fold, which conditioned a particularly marked change in the position of micro-terraces. Somewhat smaller

changes took place after the winter of 2015 – 2016 when infiltration of thawed snow water into sand

sharply increased at the beginning of March. Since the autumn of 2015, the changes in the vertical position

of terrace ledges constituted 0.12 –0.56 m on average.

## 5. Conclusions

1 The deflation processers in the northern part of the Curonian Spit were mostly affected by

frequent strong SW-W-NW winds of the maximum speed of 18 – 26 m/s, while the deflation processes

in the middle part of the spit were mostly due to E – SE – S winds of 16 – 22 m/s maximum speed. Aeolian

processes were largely influenced by long-lasting winds of the average speed exceeding 6 m/s. The

duration of such winds reached 4 thousand hours in Nida in the March – October of 2016. Strong

permanent winds induced fast sand drying and activated deflation processes, which were further

accelerated by an intensive stream of visitors to the dunes.

2. In the northern segments of the Curonian Spit (Nagliai and Vinkis dunes), aeolian microforms

(scraps, gullies, potholes, steps, small ridges and rolls) appeared in the deflation and accumulation

hollows of 35 – 40 m in absolute height. Such microforms appeared in each deflation hollow on a

windward or leeward slope of the dunes. The most intensive deflation processes took place in the hollows

of the northern slopes of the dune ridge exposed to winds blowing from the Curonian Lagoon. The

hollows on the crest of the dune ridge experienced a lower deflation degree, and those in the western

windward slope of the dune ridge suffered the lowest degree of deflation. The deflation process largely




depended on whether the hollow was closed or open. The hollows with W-E deflation passages opening
thereto underwent the most intensive transformations. W winds quickly transformed such hollows into
deflation gullies.

3. The measurements of repeated aerial photographs showed that the change in morphometric
parameters of micro-terraces was related to fluctuations in precipitation intensity. A greater infiltration of
precipitation sharply increased the weight of the sand layer and its pressure on the marl layer. In the
summer (July – August) of 2016, the amount of precipitation increased 1.6-fold compared with the
multiannual average, which gave rise to changes in micro – terraces and in the scrap of the coastal marl.
The sinking of terrace ledges reached 21%, and marl scrap height increased by 25% in some places.

4. An investigation of geodynamic processes revealed the regularities and reasons of
transformations of aeolian macro- (hollows, etc.) and micro- (micro – terraces, rolls, etc.) forms. Changes
in relief forms and their measurements confirm the "sensitivity" of aeolian sands of the Curonian Spit and
the need of regulation of their protection. Further, the abundance of visitors, their curiosity and activity
make such regulation still more relevant.

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
