# Peer review of "Intensity of geodynamic processes in the Lithuanian part of the Curonian Spit"

_Earth System Dynamics, 2017_

## Referee Comment (RC1) · Anonymous Referee #1 · 19 Jan 2017

- This paper is addressed relevant scientific questions within the scope of ESD. - The paper presents novel concepts and data. - Substantial conclusions reached are, example, "an investigation of geodynamic processes revealed the regularities and reasons of transformations of aeolian macro- (hollows, etc.) and micro- (micro – terraces, rolls, etc.) forms". - The scientific methods and assumptions valid and clearly outlined. In 76-77 line stead "tectonic subsidence" better "dune tectonic." - The results sufficient to support the interpretations and conclusions. In chapter "3.Climatic factors" says nothing about the extreme events such as storm. - The description of experiments and calculations are sufficiently complete and precise to allow their reproduction by fellow scientists. - The authors give proper credit to related work and clearly indicate their own new contribution. - The title clearly reflects the contents of the paper. - The abstract is concise and summary is complete. - The overall presentation is well structured

and clear. - The language is clear. My English is not native. - The abbreviations and units correctly defined and used. - In figure 3 of the paper must additions: 1) the cross-sections of Fig.1 and Fig.4 have to be marked are shown in Figure 3; 2) in Fig.3A unexplained black and red sings (symbols). - The number and quality of references appropriate, but in 84 line cited (Lampe et al., 2011) is not in the references. First and second references are transpose. - There is no supplementary material.

---

## Referee Comment (RC2) · Anonymous Referee #2 · 7 Feb 2017

This paper is addressed relevant scientific questions within the scope of ESD. The paper presents novel concepts, ideas and data about intensity of geodynamic processes in the Curonian Split of the Baltic Sea, and considers conditions and intensity of aeolian and dune slope transformation processes occurring in the wind-blown sand strips of the dunes. The title clearly reflects with the contents of the paper, and the abstract provide a concise and complete summary. The scientific methods and assumptions clearly outlined. In this paper mathematical formulae, symbols, abbreviations, and units correctly defined and used. The description of experiments and calculations sufficiently complete and precise to allow their reproduction by fellow scientists The research results sufficient to support the interpretations and conclusions.

Several non-essential remarks. My advice would be to shorten and clarify some of the phrases. For example, ("4.Results and discussion", 202) "According to the data

from the Nida meteorological station (wind speed and direction were measured every 3 hours ???), winds were recorded to blow for a total 4119 hours, and winds stronger than 6 m/s were recorded to blow 3966 hours (Table 1) during the warm season (period?). Would be better see in the "3.Method: part, but not in the "3.Climatic factors" part a statement (182) "To measure the local wind regime in the coastal area of the Baltic Sea, a temporary mobile meteorological station was established. It operated only in the summer of 2016."). I would recommend explaining the differentiation according strength of the wind, better. For example, what is the wind force differentiation criteria? How does this affect geomorphological processes?; and so on. It is recommended to compare the same periods, for example, why the analysis of wind period 2006-2015 (200), if the period of investigation covers the year 2016? It is recommended to restructure and clearly reason the fourth conclusion.
* * *

---

## Referee Comment (RC3) · Anonymous Referee #1 · 10 Feb 2017

I agree with the comments made by Referee 2. We will shorten and clarify thr phrases inchapters 3 and 4. The second part of fourth conclusion will be substantiated as well.

---

## Author Comment (AC2) · 31 Mar 2017

Dear reviewer.

Thank You for thorough analysis of our manuscript.

We send response for You notes: 1. Added in Fig. 3 sign of sandbound and contemporary settlements (A part) and marked cross-section places, which are in Fig. 1 and Fig. 4 (B part). 2. In line 76 changed term "dune tectonic" in "tectonic subsidence". 3. Added information about hurricane influence on dunes surface changes (185 – 190 line). 4. In References added Lampe et all., 2011. 5. Transpose the first and second position in References.

Thank You very much

Best regards

Algimantas Česnulevičius Regina MorkÅńnaitÄŮ

Please also note the supplement to this comment:
http://www.earth-syst-dynam-discuss.net/esd-2017-3/esd-2017-3-AC2-supplement.pdf

---

## Author Comment (AC3) · 31 Mar 2017

Dear reviewer.

Thank You for thorough analysis of our manuscript.

We send responses for You notes. 1. Revised and corrected wind speed measurement frequency. The wind speed measured every 6 hours (207 line). 2. Corrected summary of windy hours (207 line). 3. Transformed Table 1. The new version is clearer. 4. Part text from chapter 3, moved to chapter 2 as suggested by the reviewer (124 – 132 line). 5. Corrected the fourth conclusion.

Thank You very much

best regards

Algimantas Česnulevičius Regina MorkÅńnaitÄŮ

Please also note the supplement to this comment:
http://www.earth-syst-dynam-discuss.net/esd-2017-3/esd-2017-3-AC3-supplement.pdf

**Supplement:**

[revised manuscript text omitted]

---

## Author Response (AR1)

The answers to the RC1 reviewer comments:

1. Added in Fig. 3 sign of sandbound and contemporary settlements (A part) and marked cross-section places, which are in Fig. 1 and Fig. 4 (B part).
2. In line 76 changed term "dune tectonic" in "tectonic subsidence".
3. Added information about hurricane influence on dunes surface changes (185 – 190 line).
4. In References added Lampe et all., 2011.
5. Transpose the first and second position in References.

The answers to the RC2 reviewer comments:

1. Revised and corrected wind speed measurement frequency. The wind speed measured every 6 hours (207 line).
2. Corrected summary of windy hours (211 line).
3. Transformed Table 1. The new version is clearer.
4. Part text from chapter 3, moved to chapter 2 as suggested by the reviewer (124 – 132 line).
5. Corrected the fourth conclusion.

The answers to the RC3 reviewer comments:

1. The RC3 reviewer notes is the same as the RC2 reviewer and will have a hope that we responded to them

The answers to editor:

1. Added information and references about sea level impact to dunes slope changes during of storm.
2. Shorted 4 conclusion and not explain the human affected on sand stability.
3. Corrected mistakes in references list:
   - Berendt G.: Geologie des Kurishen Haffes und seiner Umgebung, Königsberg, 1869.
   - Corrected maps title on: Karte des Deutschen Reiches 1:100 000. K 3 II L 62. Blattnummer: 3, 8, 15, 16, 29. 1911.

---

## Author Response (AR2)

Dear Professor.

We added a fourth conclusion of article, as You suggested emphasizing the visitors impact for dune surface. Now this conclusion is as follows:

4. An investigation of geodynamic processes revealed the regularities and reasons of transformations of aeolian macro (hollows, etc.), meso- (rolls) and micro-(micro-terraces, etc.) forms. Changes in relief forms and their measurements confirm the "sensitivity" of aeolian sands of the Curonian Spit and the need of regulation of their protection. The abundance of visitors, their curiosity and activity make such regulation even more relevant. Such problematic places are as follows: Parnidis Dune, where visiting is unlimited, and Naglis Dune, where visiting is restricted and fee must be payed. In other drifting dune areas, visiting is completely banned.

With respect

Algimantas Česnulevičius